# Assessing the real-world safety of docetaxel for non-small cell lung cancer: Insights from a comprehensive analysis of FAERS data

Lei Wang[1], Hui Zhao[1], Kunpeng Yang[2], Zhe Wang[1], Chenglun Cai[1], Peiyun Lv[1], Bao Wang [1,2,3]*

**1** Jilin Cancer Hospital, Changchun, China, **2** Changchun University of Chinese Medicine, Changchun, China, **3** Tianjin Medical University, Tianjin, China

* jpch0000@163.com

## Abstract

### Background

Docetaxel is a key therapeutic agent in the treatment of non-small cell lung cancer (NSCLC), but comprehensive real-world data on its adverse effects remain scarce. This study aims to evaluate docetaxel-related adverse drug events by analyzing the Adverse Event Reporting System (FAERS) database of the U.S. Food and Drug Administration (FDA) from 2004 to 2024.

### Methods

This study utilized descriptive analysis along with four heterogeneity analysis methods: Reporting Odds Ratio (ROR), Proportional Reporting Ratio (PRR), Multivariate Gamma-Poisson Shrinker (MGPS), and Bayesian Confidence Propagation Neural Network (BCPNN), to systematically analyze 1,535 adverse event reports.

### Results

The results identified potential adverse reactions associated with docetaxel that were not included in the drug label, such as dehydration, leukopenia, acute kidney injury, and hemoptysis, in addition to well-established adverse effects. Subgroup analysis revealed that male patients were more likely to experience respiratory-related adverse events, while female patients had a higher incidence of endocrine, metabolic, and skin-related adverse events. Patients aged over 65 years exhibited an elevated risk of cardiovascular and respiratory complications.

### Conclusion

These findings offer critical evidence for the individualized monitoring and risk management of docetaxel in clinical practice, providing healthcare professionals with the

**Data availability statement:** All relevant data are included within the paper and its Supporting information files. The original datasets are publicly available through Figshare (DOI: 10.6084/m9.figshare.29624663) and the FDA Adverse Event Reporting System (FAERS) Public Dashboard [https://fis.fda.gov/extensions/FPD-QDE-FAERS/FPD-QDE-FAERS.html].

**Funding:** This work was supported by the Jilin Provincial Scientific and Technological Development Program (No. YDZJ202501ZYTS267 to B.W.). The funders had no role in study design, data collection and analysis, decision to publish, or preparation of the manuscript.

**Competing interests:** The authors have declared that no competing interests exist.

necessary information to optimize treatment regimens and ensure the safe use of docetaxel.

## 1 Introduction

Non-small cell lung cancer (NSCLC) is the most prevalent form of lung cancer worldwide, comprising approximately 85% of all lung cancer cases [1,2] and continuing to be the leading cause of cancer-related deaths [3]. Despite ongoing advancements in diagnostic and treatment methods, the prognosis for NSCLC remains poor, with the five-year survival rate remaining low [2]. According to an analysis of the Surveillance, Epidemiology, and End Results (SEER) database, the five-year survival rate for NSCLC patients (all stages) is estimated at 26.4%, while for advanced or metastatic NSCLC, the five-year survival rate remains relatively low. For instance, the SEER database reports that the five-year survival rate for stage IV NSCLC patients is estimated at 5.8%. This rate is even lower for stage IV patients aged 65 years and older, at 4.6%, compared to 7.5% for patients younger than 65 years [4]. Notably, the epidemiological characteristics of NSCLC are undergoing significant changes due to shifts in population demographics and smoking patterns, with adenocarcinoma becoming the most common and rapidly increasing subtype [3,5].

In recent years, groundbreaking progress in NSCLC treatment strategies has led to an integrated therapeutic approach combining targeted therapy, immunotherapy, and traditional chemotherapy. Within targeted therapy, EGFR-TKIs, particularly osimertinib, have proven highly effective for patients with EGFR mutations due to their exceptional central nervous system penetration and favorable safety profile [1,6]. PD-1/PD-L1 inhibitors have become the standard treatment for patients with driver gene-negative tumors, while novel immunotherapies such as antibody-drug conjugates (ADCs) and bispecific antibodies show significant promise for future applications [7,8]. For patients with locally advanced NSCLC, the adoption of multidisciplinary collaborative treatment models and perioperative immunotherapy has significantly improved disease control [3,7]. Additionally, in clinical scenarios such as immunotherapy-resistant or driver gene-negative cancers, chemotherapy agents like docetaxel continue to provide essential therapeutic benefits [7,9].

Docetaxel is a semi-synthetic derivative of paclitaxel, commonly used as part of second-line treatment regimens for non-small cell lung cancer (NSCLC), particularly following the failure of platinum-based chemotherapy (such as cisplatin or carboplatin) [10]. Additionally, it is frequently combined with other therapeutic approaches, including immune checkpoint inhibitors and targeted therapies, to enhance treatment efficacy [11]. However, the widespread use of docetaxel in combination or sequential therapies complicates the analysis of its adverse drug reactions (ADEs) [12]. The challenge arises from the difficulty in isolating and attributing ADEs specifically to docetaxel when it is used in combination with other drugs or as part of a multi-drug regimen. This often leads to confounding factors, making the identification of adverse effects directly caused by docetaxel more complex.

To minimize the impact of such confounding factors, this study excluded reports involving the co-administration of other drugs through sensitivity analysis. In this manner, we ensured a higher degree of accuracy in our data analysis, further clarifying the risks and patterns of docetaxel-related adverse reactions.

Since its FDA approval in 1996, docetaxel has established a central role in the treatment of various malignant tumors [13,14]. Docetaxel belongs to a class of chemotherapy drugs called taxanes [15,16]. This drug works by specifically binding to tubulin, preventing microtubule depolymerization, disrupting microtubule dynamics, and inducing G2/M phase cell cycle arrest and apoptosis [13,17]. Taxanes block cell growth during mitosis, affecting both tumor cells and certain normal cells, which impacts normal tissue regeneration [15,16]. These actions make docetaxel a key agent in second-line treatment for NSCLC, especially after platinum-based therapies fail [18,19]. Recent studies have demonstrated that combining docetaxel with immune checkpoint inhibitors, particularly nivolumab, provides substantial clinical benefits for NSCLC patients who have previously failed immunotherapy, leading to significant improvements in progression-free and overall survival [20,21]. Additionally, combining docetaxel with radiotherapy and targeted therapies shows potential for synergistic effects [22].

Despite its significant clinical benefits, the use of docetaxel is limited by the development of resistance and severe adverse effects, including myelosuppression, neurotoxicity, and liver dysfunction [13,14,17]. These challenges have driven ongoing research into the underlying resistance mechanisms and the development of innovative therapeutic strategies, providing new opportunities to improve clinical outcomes and advance personalized cancer treatments [23–25].

The widespread use of docetaxel in cancer treatment has resulted in an increasing number of reported adverse drug events (ADEs). However, comprehensive long-term efficacy and safety data are currently lacking, especially from large-scale real-world studies. To address this gap, this study conducted a thorough pharmacovigilance analysis using the FDA Adverse Event Reporting System (FAERS) database. By analyzing docetaxel-related adverse event data from Q1 2004 to Q1 2024, we systematically assessed its specific side effects, temporal patterns, and gender differences. The findings from this analysis will provide valuable insights for medication monitoring and risk management in clinical practice, assisting healthcare professionals in optimizing treatment regimens and ensuring the safe use of docetaxel.

## 2 Materials and methods

### 2.1 Data sources, management, and study design

This study utilized data from the FDA Adverse Event Reporting System (FAERS) database, which collects spontaneous reports from consumers (CN), pharmacists (PH), physicians (MD), other healthcare professionals (HP), registered nurses (RN), and other individuals (OT). We included all adverse event reports from Q1 2004 to Q1 2024 in which docetaxel was identified as the primary suspected drug. The data management process strictly followed the FDA's recommended procedures, focusing on two key steps: identifying and removing duplicate reports and standardizing adverse event terminology. Specifically, for reports with identical case identifiers (CASEID), the record with the most recent FDA receipt date (FDA_DT) was retained. If both CASEID and FDA_DT were identical, the report with the highest primary identifier (PRIMARYID) was kept. Adverse event terms were standardized using the MedDRA version 26.1 dictionary to ensure the accuracy of subsequent statistical analyses [26]. The detailed study process is illustrated in Fig 1. MedDRA dictionary information and the latest version can be accessed at the official website: https://www.meddra.org. This study involves human participants and adheres to local laws and institutional requirements, thus ethical approval is not required. In accordance with national laws and institutional policies, participants or their legal guardians/close relatives are not obligated to sign a written informed consent form to participate in the study.

### 2.2 Statistical analysis

This study first conducted a descriptive analysis of docetaxel-related adverse event reports and then employed four disproportionality analysis methods to assess potential adverse reaction signals: Reporting Odds Ratio (ROR), Proportional

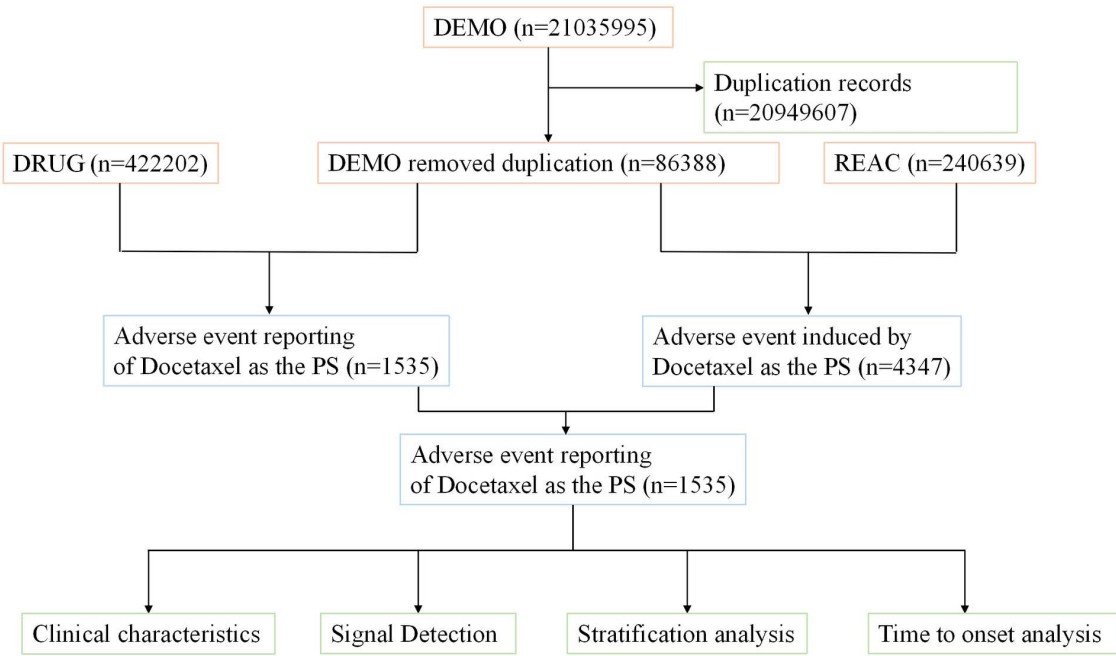

**Fig 1. Flowchart demonstrating the adverse event analysis process for Docetaxel using the FDA Adverse Event Reporting System (FAERS) database.**

Reporting Ratio (PRR), Multi-item Gamma-Poisson Shrinker (MGPS), and Bayesian Confidence Propagation Neural Network (BCPNN). An adverse event was identified as a potential adverse reaction if any of these methods met the positive determination threshold. Detailed evaluation matrices and criteria for each method are provided in S1 and S2 Tables.

The time of adverse event occurrence was defined as the interval between the initiation of docetaxel treatment (THER file record) and the onset of the adverse event (DEMO file record). Temporal trends in adverse event incidence were modeled and analyzed using the Weibull distribution. All statistical analyses were conducted using R software version 4.4.0.

## 3 Results

### 3.1 Descriptive analysis

This study analyzed 1,535 adverse event reports encompassing 4,347 adverse events, with docetaxel identified as the primary suspected drug in all cases. Regarding patient demographics, 56.4% were male and 32.5% female, with the majority (42.3%) aged between 65 and 85 years. Of the report sources, 88.9% were submitted by healthcare professionals, primarily from the United States (19.4%), Germany (16.0%), and Japan (10.2%). Detailed descriptive statistics are provided in Table 1.

### 3.2 Distribution of adverse events by system organ class (SOC)

Docetaxel-related adverse events were reported in 25 of the 27 SOC categories. As illustrated in Table 2, significant findings were noted in several categories, including Gastrointestinal Disorders, General Disorders and Administration Site Conditions, Infections and Infestations, Respiratory, Thoracic and Mediastinal Disorders, Investigations, Blood and Lymphatic System Disorders, and Neoplasms Benign, Malignant, and Unspecified (including Cysts and Polyps). The distribution of adverse events by SOC is depicted in Fig 2.

**Table 1. Clinical characteristics of Docetaxel adverse event reports from the FAERS database (Q1 2004 - Q1 2024).**

| Characteristics | Case numbers | Case proportion (%) |
|---|---|---|
| Number of events | 1535 | |
| **Gender** | | |
| Male | 866 | 56.4 |
| Female | 499 | 32.5 |
| Unknown | 170 | 11.1 |
| **Age** | | |
| Median (IQR) | 65 (58, 71) | |
| <18 | 9 | 0.6 |
| 18-65 | 568 | 37.0 |
| 65-85 | 650 | 42.3 |
| >85 | 4 | 0.3 |
| Unknown | 304 | 19.8 |
| **Top 5 Reported Countries** | | |
| United States | 298 | 19.4 |
| Germany | 246 | 16.0 |
| Japan | 157 | 10.2 |
| United Kingdom | 51 | 3.3 |
| France | 44 | 2.9 |
| **Reporter** | | |
| Healthcare professional | 1366 | 89.0 |
| Non-healthcare professional | 104 | 6.8 |
| Unknown | 65 | 4.2 |
| **Reporting year** | | |
| 2004-2008 | 464 | 30.2 |
| 2009-2013 | 191 | 12.4 |
| 2014-2018 | 317 | 20.7 |
| 2017-2024 | 563 | 36.7 |

Abbreviation: interquartile range, IQR.

### 3.3 Distribution of adverse events by preferred term (PT)

Frequency ranking and signal evaluation were performed for docetaxel-related adverse events. Among the 50 most common adverse events, known reactions included sepsis, pneumonia, neutropenia, anemia, febrile neutropenia, hypersensitivity, thrombocytopenia, anorexia, peripheral sensory neuropathy, peripheral motor neuron disease, dysgeusia, arrhythmia, hypokalemia, hypertension, and bleeding. The study also identified potential adverse reactions not listed in the labeling, such as fatigue, dehydration, decreased white blood cell count, leukopenia, acute kidney injury, haemoptysis, mucosal inflammation, hyponatremia, decreased hemoglobin, urinary tract infection, and increased gamma-glutamyltransferase. Detailed results are presented in Table 3, and all adverse events with positive signals are listed in S3 Table.

### 3.4 Subgroup analysis

Subgroup analysis of docetaxel-related adverse events revealed significant differences based on gender and age. Among the 50 most common adverse events with positive signals, those specific to males included dyspnoea, pyrexia, hypotension, decreased neutrophil count, respiratory failure, sepsis, septic shock, hypoxia, neutropenic sepsis,

**Table 2. Signal strength of Docetaxel AEs across System Organ Classes (SOC) in the FAERS database.**

| System Organ Class (SOC) | Case numbers | ROR(95%CI) | PRR(χ²) | EBGM(EBGM05) | IC(IC025) |
|---|---|---|---|---|---|
| Immune system disorders | 19 | 0.76 (0.48 - 1.2) | 0.76 (1.39) | 0.77 (0.52) | −0.38 (−1.04) |
| General disorders and administration site conditions | 584 | 0.93 (0.85 - 1.02) | 0.94 (2.56) | 0.94 (0.87) | −0.09 (−0.22) |
| Respiratory, thoracic and mediastinal disorders | 406 | 0.88 (0.79 - 0.97) | 0.89 (6.21) | 0.89 (0.82) | −0.17 (−0.32) |
| Infections and infestations* | 429 | 1.59 (1.44 - 1.76) | 1.53 (82.5) | 1.52 (1.39) | 0.6 (0.45) |
| Skin and subcutaneous tissue disorders | 211 | 0.88 (0.76 - 1.01) | 0.88 (3.34) | 0.89 (0.79) | −0.17 (−0.38) |
| Vascular disorders | 105 | 1.06 (0.87 - 1.29) | 1.06 (0.35) | 1.06 (0.9) | 0.08 (−0.21) |
| Gastrointestinal disorders* | 586 | 1.26 (1.15 - 1.37) | 1.22 (26.06) | 1.22 (1.13) | 0.28 (0.16) |
| Hepatobiliary disorders | 46 | 0.46 (0.35 - 0.62) | 0.47 (27.99) | 0.47 (0.37) | −1.08 (−1.5) |
| Neoplasms benign, malignant and unspecified (incl cysts and polyps) | 271 | 0.75 (0.66 - 0.85) | 0.77 (20.43) | 0.77 (0.7) | −0.38 (−0.56) |
| Eye disorders | 32 | 0.62 (0.43 - 0.87) | 0.62 (7.54) | 0.62 (0.46) | −0.68 (−1.19) |
| Injury, poisoning and procedural complications | 91 | 0.55 (0.44 - 0.67) | 0.56 (33) | 0.56 (0.47) | −0.83 (−1.14) |
| Metabolism and nutrition disorders* | 206 | 1.19 (1.04 - 1.38) | 1.18 (6.05) | 1.18 (1.05) | 0.24 (0.03) |
| Investigations* | 389 | 1.19 (1.07 - 1.32) | 1.17 (10.11) | 1.17 (1.07) | 0.22 (0.07) |
| Cardiac disorders | 158 | 1.13 (0.96 - 1.32) | 1.12 (2.18) | 1.12 (0.98) | 0.17 (−0.07) |
| Blood and lymphatic system disorders* | 316 | 1.48 (1.32 - 1.66) | 1.45 (44.66) | 1.43 (1.3) | 0.52 (0.35) |
| Musculoskeletal and connective tissue disorders | 94 | 0.78 (0.64 - 0.96) | 0.79 (5.5) | 0.79 (0.66) | −0.34 (−0.64) |
| Renal and urinary disorders | 67 | 0.66 (0.52 - 0.84) | 0.66 (11.45) | 0.67 (0.55) | −0.58 (−0.93) |
| Nervous system disorders | 214 | 0.92 (0.8 - 1.06) | 0.93 (1.33) | 0.93 (0.83) | −0.11 (−0.31) |
| Psychiatric disorders | 67 | 1.06 (0.83 - 1.35) | 1.06 (0.21) | 1.06 (0.86) | 0.08 (−0.28) |
| Ear and labyrinth disorders | 14 | 1.28 (0.75 - 2.18) | 1.28 (0.83) | 1.27 (0.82) | 0.35 (−0.41) |
| Endocrine disorders | 17 | 0.33 (0.2 - 0.53) | 0.33 (22.82) | 0.34 (0.23) | −1.57 (−2.25) |
| Social circumstances | 3 | 0.8 (0.26 - 2.51) | 0.8 (0.14) | 0.8 (0.31) | −0.31 (−1.77) |
| Congenital, familial and genetic disorders | 12 | 1.16 (0.66 - 2.06) | 1.16 (0.27) | 1.16 (0.72) | 0.21 (−0.6) |
| Reproductive system and breast disorders | 6 | 0.94 (0.42 - 2.1) | 0.94 (0.03) | 0.94 (0.48) | −0.09 (−1.2) |
| Surgical and medical procedures | 4 | 0.26 (0.1 - 0.7) | 0.26 (8.25) | 0.27 (0.12) | −1.91 (−3.2) |

Abbreviation: Asterisks (*) indicate statistically significant signals in algorithm; ROR, reporting odds ratio; PRR, proportional reporting ratio; EBGM, empirical Bayesian geometric mean; EBGM05, the lower limit of the 95% CI of EBGM; IC, information component; IC025, the lower limit of the 95% CI of the IC; CI, confidence interval; AEs, adverse events.

dysphagia, interstitial lung disease, pulmonary embolism, decreased hemoglobin, haemoptysis, chronic obstructive pulmonary disease, and pneumothorax. Female-specific adverse events included alopecia, hypokalemia, increased gamma-glutamyltransferase, urinary tract infection, muscular weakness, increased aspartate aminotransferase, increased blood alkaline phosphatase, psychological trauma, cerebral infarction, increased blood bilirubin, hair colour changes, hair disorder, abnormal hair texture, madarosis, cerebrovascular accident, hepatotoxicity, peripheral sensory neuropathy, hypocalcaemia, and hypereosinophilic syndrome. Detailed information is available in S4 and S5 Tables. Age-stratified analysis showed that only nine reports involved patients under 18 years old, with adverse events such as urinary tract infection and herpes zoster, which were not listed in the labeling. For patients aged 18–65, specific adverse events included fatigue, alopecia, anaemia, increased gamma-glutamyltransferase, abdominal pain, hyponatraemia, acute kidney injury, increased blood alkaline phosphatase, muscular weakness, neoplasm progression, increased aspartate aminotransferase, psychological trauma, dermatitis, increased blood bilirubin, chronic obstructive pulmonary disease, hair colour changes, hair disorder, abnormal hair texture, madarosis, abnormal hepatic function, and hepato-toxicity. Specific adverse events for patients over 65 years old included dyspnoea, respiratory failure, pleural effusion,

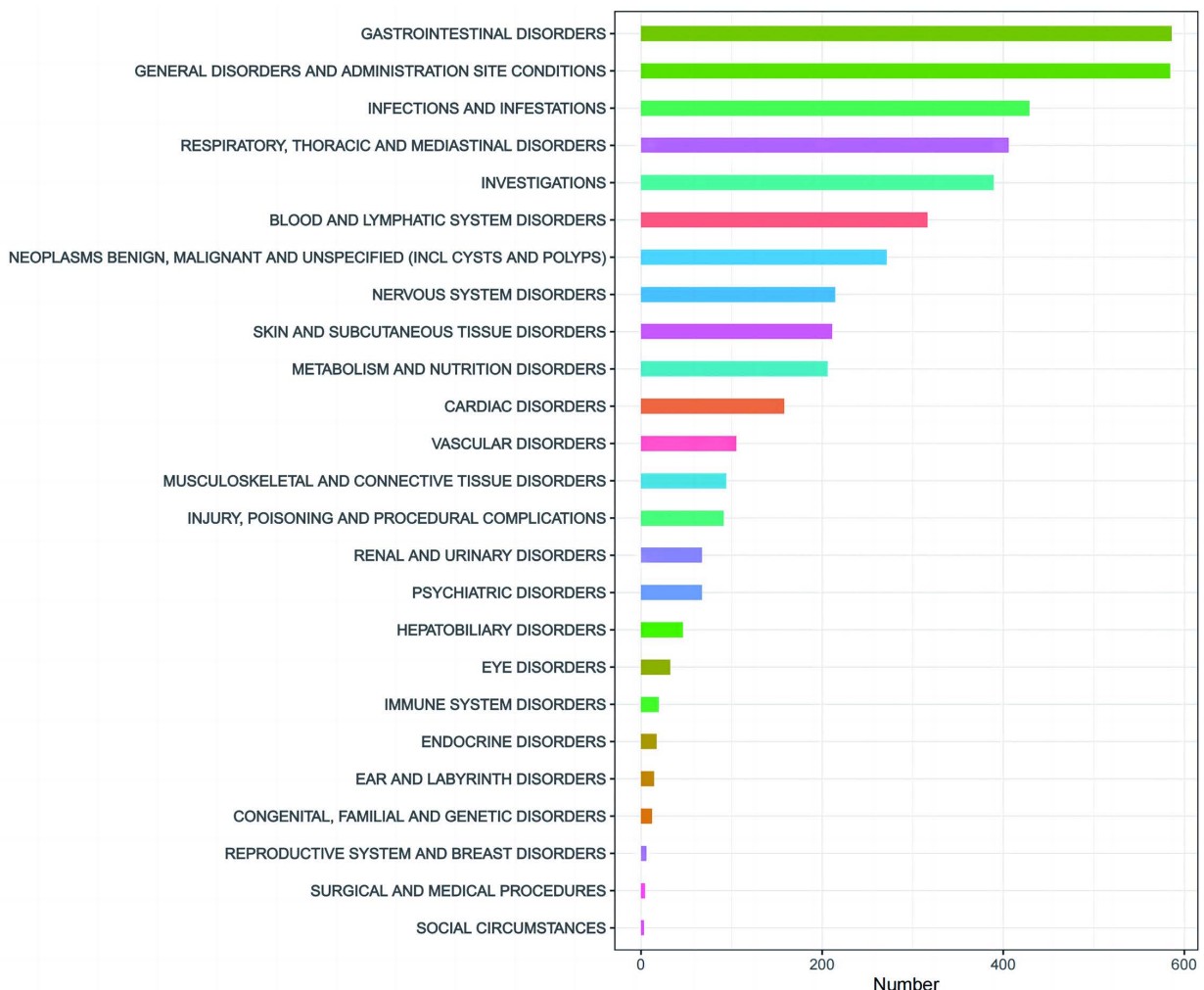

**Fig 2. Proportion of adverse events by system organ class associated with Docetaxel.** Explanation: The 'Investigations' category refers to adverse events related to abnormal laboratory test results. 'Social circumstances' refers to social factors that influence the patient's health status or reporting behavior.

hypotension, cardiac arrest, mucosal inflammation, neutropenic sepsis, septic shock, hypoxia, atrial fibrillation, polyneuropathy, dysphagia, hypereosinophilic syndrome, pneumothorax, respiratory tract infection, haemoptysis, and pulmonary embolism (see S6–S8 Tables for details).

## 3.5 Sensitivity analysis

Docetaxel is often used in combination with other drugs, such as carboplatin and cisplatin. After excluding reports involving the concurrent use of other medications, 1,199 reports encompassing 1,701 adverse events were identified. Persistent potential adverse reactions included diarrhoea, pneumonia, febrile neutropenia, dyspnoea, vomiting, dehydration, respiratory failure, leukopenia, hypotension, haemoptysis, hyponatraemia, septic shock, abdominal pain, decreased hemoglobin, hypokalemia, cardiac arrest, urinary tract infection, atrial fibrillation, increased gamma-glutamyltransferase, confusional state, neutropenic sepsis, and hypoxia (see S9 Table for details).

**Table 3. Top 50 frequency of adverse events at the PT level for Docetaxel.**

| PT | Case numbers | ROR(95%CI) | PRR($\chi^2$) | EBGM(EBGM05) | IC(IC025) |
|---|---|---|---|---|---|
| Diarrhoea* | 200 | 2 (1.73 - 2.31) | 1.95 (91.94) | 1.92 (1.7) | 0.94 (0.73) |
| Malignant neoplasm progression | 110 | 0.71 (0.59 - 0.86) | 0.72 (12.62) | 0.72 (0.61) | −0.47 (−0.75) |
| Pneumonia* | 106 | 1.71 (1.41 - 2.08) | 1.7 (29.83) | 1.68 (1.42) | 0.74 (0.46) |
| Nausea* | 106 | 1.65 (1.36 - 2.01) | 1.64 (25.84) | 1.62 (1.37) | 0.69 (0.41) |
| Death | 87 | 0.61 (0.49 - 0.76) | 0.62 (20.84) | 0.62 (0.52) | −0.68 (−0.99) |
| Fatigue* | 85 | 1.47 (1.19 - 1.83) | 1.46 (12.34) | 1.45 (1.21) | 0.54 (0.22) |
| Febrile neutropenia* | 85 | 2.99 (2.4 - 3.73) | 2.95 (104.94) | 2.85 (2.37) | 1.51 (1.19) |
| Pyrexia | 66 | 1.24 (0.97 - 1.59) | 1.24 (3.01) | 1.23 (1) | 0.3 (−0.06) |
| Vomiting* | 66 | 1.36 (1.07 - 1.74) | 1.36 (6.16) | 1.35 (1.1) | 0.43 (0.07) |
| Neutropenia* | 65 | 2.05 (1.6 - 2.63) | 2.04 (33.29) | 2 (1.62) | 1 (0.63) |
| Dyspnoea | 60 | 0.94 (0.73 - 1.22) | 0.94 (0.22) | 0.94 (0.76) | −0.09 (−0.46) |
| Dehydration* | 54 | 1.68 (1.28 - 2.2) | 1.67 (14.2) | 1.65 (1.31) | 0.72 (0.33) |
| White blood cell count decreased* | 54 | 3.26 (2.47 - 4.29) | 3.23 (78.8) | 3.1 (2.46) | 1.63 (1.23) |
| Disease progression | 50 | 1.48 (1.12 - 1.96) | 1.48 (7.51) | 1.46 (1.15) | 0.55 (0.14) |
| General physical health deterioration* | 49 | 1.94 (1.45 - 2.58) | 1.93 (21.21) | 1.89 (1.49) | 0.92 (0.5) |
| Anaemia | 48 | 1 (0.75 - 1.33) | 1 (0) | 1 (0.79) | 0 (−0.42) |
| Non-small cell lung cancer | 46 | 1 (0.75 - 1.34) | 1 (0) | 1 (0.78) | 0 (−0.42) |
| Leukopenia* | 46 | 3.04 (2.26 - 4.1) | 3.02 (59.15) | 2.91 (2.27) | 1.54 (1.11) |
| Alopecia* | 45 | 3.92 (2.9 - 5.32) | 3.89 (90.56) | 3.7 (2.87) | 1.89 (1.45) |
| Decreased appetite | 43 | 0.97 (0.71 - 1.31) | 0.97 (0.05) | 0.97 (0.75) | −0.05 (−0.49) |
| Asthenia | 38 | 0.99 (0.72 - 1.37) | 0.99 (0) | 0.99 (0.76) | −0.01 (−0.48) |
| Stomatitis* | 36 | 2.16 (1.54 - 3.01) | 2.15 (21.29) | 2.1 (1.59) | 1.07 (0.59) |
| Neutrophil count decreased* | 34 | 2.21 (1.57 - 3.12) | 2.2 (21.43) | 2.15 (1.61) | 1.11 (0.61) |
| Hypotension* | 33 | 2.2 (1.55 - 3.12) | 2.19 (20.56) | 2.14 (1.6) | 1.1 (0.59) |
| Respiratory failure | 33 | 1.25 (0.89 - 1.77) | 1.25 (1.63) | 1.24 (0.93) | 0.32 (−0.19) |
| Pneumonitis | 32 | 0.81 (0.57 - 1.15) | 0.81 (1.37) | 0.82 (0.61) | −0.29 (−0.8) |
| Pleural effusion | 31 | 0.95 (0.67 - 1.36) | 0.96 (0.06) | 0.96 (0.71) | −0.07 (−0.58) |
| Sepsis* | 29 | 1.46 (1.01 - 2.11) | 1.45 (4.01) | 1.44 (1.06) | 0.53 (−0.01) |
| Acute kidney injury* | 28 | 1.57 (1.08 - 2.29) | 1.57 (5.65) | 1.55 (1.13) | 0.64 (0.09) |
| Haemoptysis* | 28 | 1.5 (1.03 - 2.19) | 1.5 (4.58) | 1.49 (1.09) | 0.57 (0.03) |
| Cardiac arrest* | 27 | 4.39 (2.97 - 6.51) | 4.37 (65.1) | 4.12 (2.97) | 2.04 (1.48) |
| Mucosal inflammation* | 27 | 2.45 (1.66 - 3.61) | 2.44 (22.05) | 2.38 (1.72) | 1.25 (0.69) |
| Hyponatraemia* | 25 | 1.85 (1.24 - 2.76) | 1.84 (9.35) | 1.82 (1.3) | 0.86 (0.28) |
| Neoplasm progression | 24 | 1.32 (0.88 - 1.98) | 1.32 (1.83) | 1.31 (0.93) | 0.39 (−0.19) |
| Drug ineffective | 23 | 1.06 (0.7 - 1.6) | 1.06 (0.07) | 1.06 (0.75) | 0.08 (−0.52) |
| Haemoglobin decreased* | 23 | 1.78 (1.18 - 2.71) | 1.78 (7.64) | 1.76 (1.24) | 0.81 (0.21) |
| Septic shock* | 22 | 2.96 (1.93 - 4.56) | 2.95 (27.03) | 2.85 (1.99) | 1.51 (0.89) |
| Abdominal pain | 22 | 1.33 (0.87 - 2.03) | 1.33 (1.74) | 1.32 (0.93) | 0.4 (−0.21) |
| Hypokalaemia* | 22 | 1.91 (1.25 - 2.92) | 1.9 (9.14) | 1.87 (1.31) | 0.91 (0.29) |
| Urinary tract infection* | 22 | 2.22 (1.45 - 3.41) | 2.22 (14.18) | 2.17 (1.52) | 1.12 (0.5) |
| Gamma-glutamyltransferase increased* | 21 | 3.29 (2.11 - 5.11) | 3.27 (31.33) | 3.14 (2.17) | 1.65 (1.02) |
| Cough | 20 | 0.83 (0.53 - 1.29) | 0.83 (0.72) | 0.83 (0.57) | −0.27 (−0.91) |
| Atrial fibrillation* | 20 | 1.63 (1.04 - 2.55) | 1.63 (4.7) | 1.61 (1.11) | 0.69 (0.05) |
| Neutropenic sepsis* | 19 | 3.92 (2.46 - 6.25) | 3.9 (38.35) | 3.71 (2.51) | 1.89 (1.22) |
| Hypoxia* | 18 | 1.74 (1.09 - 2.78) | 1.73 (5.44) | 1.71 (1.16) | 0.78 (0.1) |

*(Continued)*

**Table 3.** (Continued)

| PT | Case numbers | ROR(95%CI) | PRR(χ²) | EBGM(EBGM05) | IC(IC025) |
|---|---|---|---|---|---|
| Muscular weakness* | 18 | 2.11 (1.32 - 3.39) | 2.11 (10.13) | 2.07 (1.39) | 1.05 (0.37) |
| Pulmonary embolism | 17 | 0.62 (0.39 - 1.01) | 0.62 (3.81) | 0.63 (0.42) | −0.67 (−1.35) |
| Dysphagia | 17 | 1.28 (0.79 - 2.07) | 1.28 (1) | 1.27 (0.85) | 0.35 (−0.34) |
| Myalgia* | 16 | 1.72 (1.04 - 2.83) | 1.72 (4.64) | 1.69 (1.12) | 0.76 (0.05) |
| Interstitial lung disease | 16 | 0.37 (0.22 - 0.6) | 0.37 (17.36) | 0.37 (0.25) | −1.42 (−2.13) |

Abbreviation: Asterisks (*) indicate statistically significant signals in algorithm; ROR, reporting odds ratio; PRR, proportional reporting ratio; EBGM, empirical Bayesian geometric mean; EBGM05, the lower limit of the 95% CI of EBGM; IC, information component; IC025, the lower limit of the 95% CI of the IC; CI, confidence interval; PT, preferred term.

### 3.6 Adverse event occurrence time and weibull distribution analysis

Time distribution analysis indicated that docetaxel-related adverse events primarily occurred within 30 days of administration. This early failure pattern was confirmed through Weibull distribution analysis. The specific time distribution and cumulative incidence rates are presented in Figs 3 and 4, with detailed analysis parameters available in Table 4.

## 4 Discussion

This study offers a comprehensive assessment of docetaxel-related adverse events by analyzing data from the FDA Adverse Event Reporting System (FAERS) since 2004. We confirmed known adverse reactions listed in the drug's labeling, including sepsis, pneumonia, neutropenia, anemia, febrile neutropenia, hypersensitivity reactions, thrombocytopenia, anorexia, peripheral sensory neuropathy, peripheral motor neuropathy, taste disturbances, arrhythmias, hypokalemia, hypertension, and bleeding. Additionally, we identified several adverse events not listed in the labeling, such as metabolism and nutrition disorders (e.g., dehydration, hyponatremia), blood and lymphatic system disorders (e.g., leukopenia), renal and urinary disorders (e.g., acute kidney injury, urinary tract infection), respiratory, thoracic and mediastinal disorders (e.g., haemoptysis), gastrointestinal disorders (e.g., mucosal inflammation), and investigations (e.g., decreased white blood cell count, decreased hemoglobin, increased gamma-glutamyltransferase). Our findings highlight the ultimate importance of enhancing monitoring, especially within the first 30 days of treatment, to promptly identify and manage potential adverse reactions.

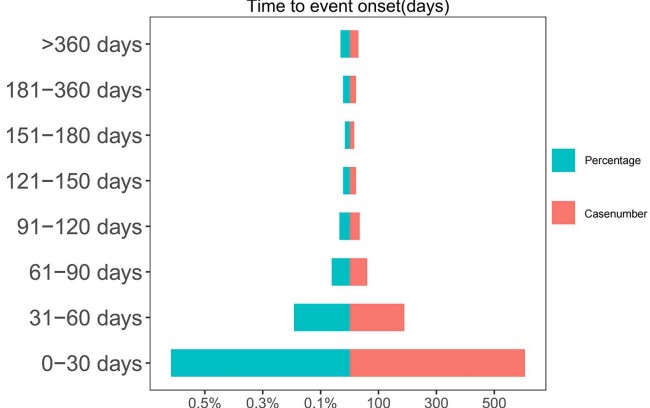

**Fig 3. Time to onset of Docetaxel-induced adverse events.**

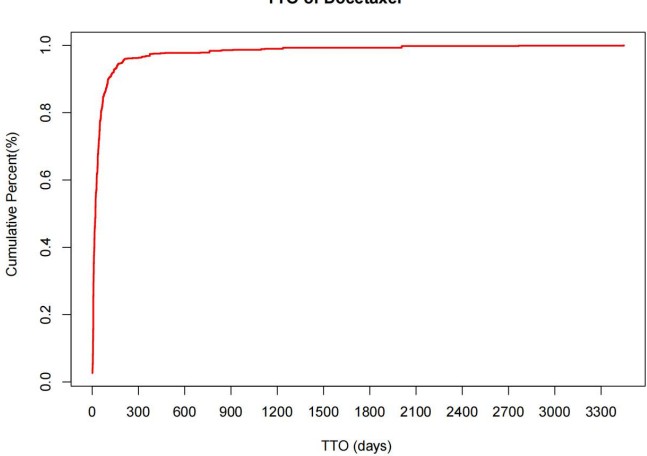

**TTO of Docetaxel**

*Fig 4. Cumulative incidence of Docetaxel-related adverse events over time.*

**Table 4. Time to onset of Docetaxel-associated adverse events and Weibull distribution analysis.**

| Drug | TTO(days) | | Weibull distribution | | Type |
|---|---|---|---|---|---|
| | Case reports | Median(d)(IQR) | Scale parameter: α(95%CI) | Shape parameter: β(95%CI) | |
| Docetaxel | 982 | 21(7-49) | 42.73(38.35-47.11) | 0.65(0.62-0.68) | Early failure |

Abbreviation: TTO,time to onset; CI, confidence interval; IQR, interquartile range.

The association between docetaxel treatment and dehydration in NSCLC patients is particularly concerning. Our FAERS findings are supported by several key clinical trials. In the JAVELIN Lung 200 phase III study comparing docetaxel with avelumab, dehydration was reported as a serious adverse event in the docetaxel group, resulting in one treatment-related death [27]. The risk of dehydration appears to increase when docetaxel is combined with other drugs [28]. Several factors may contribute to docetaxel-induced dehydration: firstly, docetaxel can cause gastrointestinal toxicity, such as diarrhea, which was reported as a grade 3–5 adverse event in the JAVELIN trial, leading to fluid loss and electrolyte imbalance [27]. Secondly, as noted by Chouaid et al., anorexia and reduced oral intake associated with docetaxel may further exacerbate dehydration risk [29]. Therefore, healthcare providers should implement strict monitoring protocols to detect early signs of dehydration, particularly in elderly patients and those undergoing combination therapy. Preventive measures, such as adequate hydration support and patient education about fluid intake, may help mitigate this risk. Additionally, the economic impact of managing dehydration, which often requires hospitalization and supportive care, significantly increases the overall cost of managing adverse events in NSCLC treatment [29]. This analysis delves into the necessity of balancing docetaxel's therapeutic benefits with the risk of dehydration-related complications, guiding clinical decision-making. Future studies should explore factors that predict susceptibility to docetaxel-induced dehydration and develop targeted prevention strategies.

Another significant adverse event is acute kidney injury. Our findings align with the JAVELIN Lung 200 study, which reported treatment-related deaths due to renal dysfunction in the docetaxel group [28]. Santana-Davila et al. further confirmed a substantial risk of acute kidney injury and dehydration-related complications in platinum-based chemotherapy regimens (29.2% vs. 15.5%, $p < 0.01$) [30]. The mechanisms underlying docetaxel-induced renal injury may include direct tubular toxicity, chemotherapy-induced dehydration, and individual susceptibility factors [31]. Consequently, it is

recommended to assess renal function before treatment, monitor renal parameters regularly during therapy, provide timely hydration, and consider individualized dose adjustments for high-risk patients. These findings underscore the ultimate need for vigilant renal function monitoring and management during docetaxel treatment, and clinicians should be aware of the potential for acute kidney injury.

Hemoptysis during docetaxel treatment is a potentially fatal adverse event that is not adequately addressed in the drug's labeling, significantly impacting its clinical application. Hemoptysis, as a life-threatening symptom, not only causes severe physical damage but also leads to considerable psychological stress and can jeopardize patient safety, potentially resulting in treatment discontinuation. Therefore, establishing a stringent monitoring protocol is essential, including regular evaluation of respiratory symptoms and bleeding risks. For patients experiencing hemoptysis, immediate hemostatic measures and airway management should be implemented, and treatment should be adjusted or discontinued if necessary to ensure patient safety. This proactive approach can effectively control symptoms and reduce the risk of fatal complications.

Our study also identified increased gamma-glutamyltransferase (GGT) as a significant adverse event in NSCLC patients treated with docetaxel. Liver dysfunction is common with chemotherapy drugs, and elevated GGT, an indicator of liver injury, requires particular attention. In the ASCEND-5 study, the incidence of GGT elevation in the chemotherapy group (including docetaxel) was 1% [32]. GGT elevation may reflect hepatocellular damage, which could compromise treatment safety and necessitate treatment adjustments. Such liver dysfunction can affect drug metabolism and increase the risk of other complications. Given that GGT elevation is an important marker to monitor during docetaxel treatment, regular liver function monitoring is essential for NSCLC patients receiving docetaxel therapy [32]. This includes baseline liver evaluation before treatment, regular monitoring during therapy, and prompt adjustments if abnormalities arise. Proactively managing this adverse reaction through structured monitoring, timely interventions, and individualized dose adjustments is crucial for ensuring treatment safety and maximizing patient benefit.

The subgroup analysis of our study indicated that additional attention should be given to respiratory symptoms, such as dyspnea, respiratory failure, and interstitial lung disease, as well as infection-related complications like sepsis and neutropenic sepsis in male patients. In female patients, particular caution should be exercised regarding liver function abnormalities (e.g., elevated GGT, AST) and hair-related adverse reactions (e.g., alopecia, hair color changes). Notably, hair-related adverse events can cause significant psychological distress, especially in female patients [32,33].

Regarding age stratification, there were only nine reports for patients under 18, mostly involving off-label adverse events like urinary tract infections and herpes zoster. In the 18–65 age group, the most common adverse events were fatigue, alopecia, and anemia, which can severely impact quality of life [34]. In patients over 65, cardiovascular and respiratory complications, such as atrial fibrillation, respiratory failure, and pulmonary embolism, should be closely monitored, as these may increase the risk of treatment-related mortality [35,36]. These age- and sex-specific adverse event patterns are crucial for tailoring individualized monitoring strategies and optimizing treatment plans. Clinicians should adopt appropriate preventive measures and monitoring plans based on the patient's demographic characteristics when using docetaxel for treatment [37].

This study has several limitations. First, the FAERS database, a voluntary reporting system used by consumers, doctors, and pharmacists, may contain incomplete or inaccurate data. For example, data on drug exposure in patients is not available. Second, the data analyzed in this study are limited, and larger datasets are needed to validate our findings. However, by focusing on both the drug itself and its indications, we have enhanced the specificity of the results. While disproportionality analysis effectively identifies positive signals for adverse events, it does not establish causality between docetaxel and these events. Future long-term prospective studies are needed to confirm the potential adverse reactions identified in this study.

## 5 Conclusion

This study offers a comprehensive analysis of docetaxel-related adverse events by utilizing data from the FAERS database, focusing on reports from the first quarter of 2004 to the first quarter of 2024. Our findings confirm the known adverse reactions listed in the drug's labeling, including sepsis, pneumonia, neutropenia, anemia, febrile neutropenia, and hypersensitivity reactions. Additionally, we identified several potential adverse events not mentioned in the labeling, such as fatigue, dehydration, leukopenia, acute kidney injury, hemoptysis, mucosal inflammation, hyponatremia, and elevated gamma-glutamyltransferase.

Subgroup analysis by sex and age revealed distinct patterns of adverse events across different patient groups. Male patients were more likely to experience respiratory-related adverse events, such as dyspnea, respiratory failure, and interstitial lung disease, while female patients exhibited a higher incidence of endocrine, metabolic, and skin-related adverse events, including alopecia, hypokalemia, and liver toxicity. Among patients over 65, cardiovascular and respiratory complications, such as atrial fibrillation, respiratory failure, and pleural effusion, were more prevalent.

These findings offer critical safety information for clinicians, emphasizing the importance of personalized monitoring and management based on patient characteristics, such as sex and age, during docetaxel treatment. Additionally, the data serve as a valuable foundation for optimizing the use of docetaxel in the treatment of non-small cell lung cancer, ultimately enhancing both treatment safety and efficacy.

## Supporting information

**S1 Table. Two-by-two contingency table for disproportionality analyses.**
(DOCX)

**S2 Table. Four major algorithms used for signal detection.**
(DOCX)

**S3 Table. All adverse events meeting the positive signal threshold at the PT level from FAERS data.**
(DOCX)

**S4 Table. Top 50 most frequent adverse events for Docetaxel at the preferred term (PT) level in males from FAERS data.**
(DOCX)

**S5 Table. Top 50 most frequent adverse events for Docetaxel at the PT level in females from FAERS data.**
(DOCX)

**S6 Table. Adverse events at the PT level for Docetaxel in patients aged under 18 from FAERS data.**
(DOCX)

**S7 Table. Top 50 most frequent adverse events for Docetaxel at the PT level in patients aged 18–65 from FAERS data.**
(DOCX)

**S8 Table. Top 50 most frequent adverse events for Docetaxel at the PT level in patients aged over 65 from FAERS data.**
(DOCX)

**S9 Table. Top 50 most frequent adverse events for Docetaxel excluding common medication co-usage at the PT level from FAERS data.**
(DOCX)

## Author contributions

**Conceptualization:** Hui Zhao.

**Formal analysis:** Kunpeng Yang.

**Funding acquisition:** Bao Wang.

**Investigation:** Zhe Wang.

**Methodology:** Lei Wang.

**Project administration:** Bao Wang.

**Software:** Chenglun Cai.

**Supervision:** Bao Wang.

**Validation:** Peiyun Lv.

**Visualization:** Kunpeng Yang.

**Writing – original draft:** Lei Wang.

**Writing – review & editing:** Bao Wang.

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
