## [Decision Letter · Decision Letter 0]

1 Jun 2025

PONE-D-25-01841Assessing the real-world safety of Docetaxel for non-small cell lung cancer: insights from a comprehensive analysis of FAERS dataPLOS ONE

Dear Dr. Wang ,

Thank you for submitting your manuscript to PLOS ONE. After careful consideration, we feel that it has merit but does not fully meet PLOS ONE’s publication criteria as it currently stands. Therefore, we invite you to submit a revised version of the manuscript that addresses the points raised during the review process.

We look forward to receiving your revised manuscript.

Kind regards,

Guocan Yu

Academic Editor

PLOS ONE

Journal Requirements:

“Jilin Provincial Scientific and Technological Development Program (No. 20210203186SF;YDZJ202501ZYTS267).”

5. Please include your tables as part of your main manuscript and remove the individual files. Please note that supplementary tables (should remain/ be uploaded) as separate "supporting information" files.

Reviewers' comments:

Reviewer's Responses to Questions

**Comments to the Author**

1. Is the manuscript technically sound, and do the data support the conclusions?

Reviewer #1: Partly

Reviewer #2: Yes

2. Has the statistical analysis been performed appropriately and rigorously? 

Reviewer #1: Yes

Reviewer #2: Yes

3. Have the authors made all data underlying the findings in their manuscript fully available?

Reviewer #1: Yes

Reviewer #2: Yes

4. Is the manuscript presented in an intelligible fashion and written in standard English?

Reviewer #1: Yes

Reviewer #2: Yes

5. Review Comments to the Author

Reviewer #1: This is a well-written study that examined the real-world safety of docetaxel--a common chemotherapy agent used to treat several types of solid tumors--here specifically in non-small cell lung cancer (NSCLC) patients. The authors leveraged adverse drug event (ADE) data from the publicly accessible FAERS from 2004 to 2024. Using several statistical methods, the authors identified multiple known ADEs but also several unlabeled events as well as age and gender differences in stratified analyses. The findings provide useful summary data to inform healthcare providers and patients regarding risk management in the use of docetaxel for NSCLC therapy. Nevertheless, the overall presentation could be significantly improved, and I have the following specific comments and concerns:

1. In the Introduction, the authors mentioned the five-year survival rate of NSCLC remains low. It would be more informative to provide actual number and source (e.g. 5-yr survival rate of late-stage NSCLC based on American Cancer Society’s Cancer Statistics (US) or WHO/IARC global survey). This allows readers to assess the updated information in context, especially in view of the recent improvement in lung cancer survival rates in some developed countries.

2. Regarding information on docetaxel, please briefly add the fact that it belongs to a general class of chemotherapy drugs (taxanes) that block cellular growth during mitosis, thus affecting both normal and tumor cells.

3. Since docetaxel is used primarily as a second-line drug or in combination therapy for NSCLC that has metastasized, it is sometimes difficult to tease out the ADEs of multiple treatments, even if docetaxel is reported as the primary suspected drug in the FAERS. Are data on first-line or combination treatment available so that investigators can sift through the cases to screen out confounding factors?

4. In the Methods section on statistical analysis, it is stated that an adverse event was identified as potential adverse reaction if any of the ROR, PRR, MGPS or BCPNN analysis methods met the positive determination threshold (as listed in Supplementary Tables 2). For example, the threshold criteria for ROR are the lower limit of 95% CI must be greater than 1 and number of events greater or equal to 3. However, when looking at the adverse events meeting the positive signal threshold (Supplementary Tables 3), I find some selected ADEs do not meet any of the positive threshold criteria. Again, as an example, the PT “malignant neoplasm progression” has ROR lower 95% CI=0.59 (not >1); PRR= -0.72 (not>=2); EBGM05=0.61 (not >2); IC025= -0.75 (not >0). Please explain and justify the threshold criteria. If there are any errors in this table, please correct.

5. It would be helpful to provide a website address or literature reference for the MedDRA dictionary mentioned in the Methods section.

6. In the study design (Figure 1), it shows that there are 1535 ADEs reporting docetaxel as the primary suspected drug and 4347 events of docetaxel induced ADEs. Please clarify the differences between the two searches and why only the former was included in the final analysis.

7. It seems the figure legends for Figures 1 to 3 are missing from the main manuscript. Also, Figure 3 is not cited in the text. Please add the missing information.

8. Please enlarge the font size and graph for Figure 2. It is difficult to view the variable labels as presented.

9. For Table 1, change the label for missing data from “Miss” to “Unknown.”

10. Please reformat Supplementary Tables 3 to 9 in the Supplementary Material file to ensure the table columns are wide enough to accommodate the numbers to prevent wrapping and allow clarity in viewing.

Reviewer #2: This is a very interesting paper addressing the adverse effects of docetaxel. It presents a well-executed description of the findings, with an appropriate acknowledgment of the study’s limitations.

For the sake of improved clarity and readability, I would suggest the following:

Figure 1 could be refined, as it currently includes some dashes and symbols that seem out of place or may confuse the reader.

In the adverse effects graph, the categories "Investigations" and "Social circumstances" are unclear. It would be helpful to provide a brief explanation of what these terms refer to in this context.

6. PLOS authors have the option to publish the peer review history of their article (what does this mean? ). If published, this will include your full peer review and any attached files.

**Do you want your identity to be public for this peer review?** For information about this choice, including consent withdrawal, please see our Privacy Policy .

Reviewer #1: **Yes: ** David W Chang

Reviewer #2: **Yes: ** Borja Aguinagalde, PhD, MD

---

## [Author Response · Author response to Decision Letter 1]

23 Jul 2025

Reviewer 1

1. In the Introduction, the authors mentioned the five-year survival rate of NSCLC remains low. It would be more informative to provide actual number and source (e.g. 5-yr survival rate of late-stage NSCLC based on American Cancer Society’s Cancer Statistics (US) or WHO/IARC global survey).

Response:

Thank you very much for your thoughtful comment and valuable suggestion regarding the inclusion of specific survival data in the Introduction. We truly appreciate your input, and, as you recommended, we have updated the manuscript to provide more precise figures on the five-year survival rate for non-small cell lung cancer (NSCLC). We have incorporated data from the Surveillance, Epidemiology, and End Results (SEER) database, which offers more detailed insights into the survival rates for NSCLC patients at various stages, including advanced or metastatic disease.

We hope that the revised text provides more context and up-to-date information, especially in light of recent improvements in survival rates in some developed countries.

Revised Text:

"Non-small cell lung cancer (NSCLC) is the most prevalent form of lung cancer worldwide, comprising approximately 85% of all lung cancer cases [1, 2] and continuing to be the leading cause of cancer-related deaths [3]. Despite ongoing advancements in diagnostic and treatment methods, the prognosis for NSCLC remains poor, with the five-year survival rate remaining low [2]. According to an analysis of the Surveillance, Epidemiology, and End Results (SEER) database, the five-year survival rate for NSCLC patients (all stages) is estimated at 26.4%, while for advanced or metastatic NSCLC, the five-year survival rate remains relatively low. For instance, the SEER database reports that the five-year survival rate for stage IV NSCLC patients is estimated at 5.8%. This rate is even lower for stage IV patients aged 65 years and older, at 4.6%, compared to 7.5% for patients younger than 65 years [4]. Notably, the epidemiological characteristics of NSCLC are undergoing significant changes due to shifts in population demographics and smoking patterns, with adenocarcinoma becoming the most common and rapidly increasing subtype [3, 5]."

We hope this revision addresses your concern and enhances the manuscript's clarity and informational value.

2. Regarding information on docetaxel, please briefly add the fact that it belongs to a general class of chemotherapy drugs (taxanes) that block cellular growth during mitosis, thus affecting both normal and tumor cells.

Response:

We sincerely appreciate your thoughtful suggestion regarding the classification of docetaxel. As per your recommendation, we have revised the manuscript to clarify that docetaxel belongs to the taxane class of chemotherapy agents, which block cellular growth during mitosis. We have also highlighted how this mechanism of action affects both tumor cells and certain normal cells, which impacts normal tissue regeneration. We hope that this addition improves the understanding of docetaxel’s broader clinical implications.

Revised Text:

"Since its FDA approval in 1996, docetaxel has established a central role in the treatment of various malignant tumors [13, 14]. Docetaxel belongs to a class of chemotherapy drugs called taxanes [15, 16]. This drug works by specifically binding to tubulin, preventing microtubule depolymerization, disrupting microtubule dynamics, and inducing G2/M phase cell cycle arrest and apoptosis [13, 17]. Taxanes block cell growth during mitosis, affecting both tumor cells and certain normal cells, which impacts normal tissue regeneration [15, 16]. These actions make docetaxel a key agent in second-line treatment for NSCLC, especially after platinum-based therapies fail [18, 19]. Recent studies have demonstrated that combining docetaxel with immune checkpoint inhibitors, particularly nivolumab, provides substantial clinical benefits for NSCLC patients who have previously failed immunotherapy, leading to significant improvements in progression-free and overall survival [20, 21]. Additionally, combining docetaxel with radiotherapy and targeted therapies shows potential for synergistic effects [22]."

We hope this revision addresses your comment and enhances the clarity and completeness of the manuscript.

3. Since docetaxel is used primarily as a second-line drug or in combination therapy for NSCLC that has metastasized, it is sometimes difficult to tease out the ADEs of multiple treatments, even if docetaxel is reported as the primary suspected drug in the FAERS. Are data on first-line or combination treatment available so that investigators can sift through the cases to screen out confounding factors?

Response:

Thank you for your insightful and thoughtful comment regarding the challenges of isolating adverse drug events (ADEs) related to docetaxel, particularly in combination therapies or multi-drug regimens. We completely agree that this is a complex issue. In the revised manuscript, we have clarified that while docetaxel is widely used as a second-line treatment following platinum-based chemotherapy failure, its use in combination with other therapies does present challenges in distinguishing the specific ADEs caused by docetaxel alone.

To minimize the impact of confounding factors, we have conducted a sensitivity analysis that excluded reports involving the co-administration of other drugs. We hope this approach improves the accuracy of our data analysis and better clarifies the risks and patterns associated specifically with docetaxel-related ADEs.

Revised Text:

"Docetaxel is a semi-synthetic derivative of paclitaxel, commonly used as part of second-line treatment regimens for non-small cell lung cancer (NSCLC), particularly following the failure of platinum-based chemotherapy (such as cisplatin or carboplatin) [10]. Additionally, it is frequently combined with other therapeutic approaches, including immune checkpoint inhibitors and targeted therapies, to enhance treatment efficacy [11]. However, the widespread use of docetaxel in combination or sequential therapies complicates the analysis of its adverse drug reactions (ADEs) [12]. The challenge arises from the difficulty in isolating and attributing ADEs specifically to docetaxel when it is used in combination with other drugs or as part of a multi-drug regimen. This often leads to confounding factors, making the identification of adverse effects directly caused by docetaxel more complex."

To minimize the impact of such confounding factors, this study excluded reports involving the co-administration of other drugs through sensitivity analysis. In this manner, we ensured a higher degree of accuracy in our data analysis, further clarifying the risks and patterns of docetaxel-related adverse reactions."

We hope that this additional explanation clarifies the steps we have taken to address the issue of confounding factors in our analysis and improves the overall quality of the manuscript.

4. In the Methods section on statistical analysis, it is stated that an adverse event was identified as a potential adverse reaction if any of the ROR, PRR, MGPS or BCPNN analysis methods met the positive determination threshold (as listed in Supplementary Tables 2). For example, the threshold criteria for ROR are the lower limit of 95% CI must be greater than 1 and number of events greater or equal to 3. However, when looking at the adverse events meeting the positive signal threshold (Supplementary Tables 3), I find some selected ADEs do not meet any of the positive threshold criteria. Again, as an example, the PT “malignant neoplasm progression” has ROR lower 95% CI=0.59 (not >1); PRR= -0.72 (not>=2); EBGM05=0.61 (not >2); IC025= -0.75 (not >0). Please explain and justify the threshold criteria. If there are any errors in this table, please correct.

Response:

Thank you very much for your careful review and thoughtful comment. We appreciate your attention to the discrepancies between the adverse events listed in Supplementary Table 3 and the threshold criteria for statistical methods. After reviewing your comment, we can confirm that there was no error in the statistical analysis itself. However, we did identify that an incorrect version of the table was uploaded. Initially, we included both negative and positive signals in the table.

In response to your comment, we have now updated the table to only reflect those adverse events that meet the positive signal criteria as outlined in the Methods section. This ensures that the table is aligned with the threshold criteria for ROR, PRR, MGPS, and BCPNN analysis methods. We believe this change improves the clarity of the data presentation.

We hope this correction addresses your concern and provides a clearer view of the findings.

5. It would be helpful to provide a website address or literature reference for the MedDRA dictionary mentioned in the Methods section.

Response:

Thank you for your valuable suggestion regarding the inclusion of a reference or website address for the MedDRA dictionary. As you recommended, we have updated the manuscript to provide the official website for the MedDRA dictionary, which can be accessed for further information and the latest version. We have also added the relevant citation (reference 26) in the Methods section.

Revised Text:

"MedDRA dictionary information and the latest version can be accessed at the official website: [https://www.meddra.org](https://www.meddra.org)."

We hope this addition will help provide readers with a clearer reference for the MedDRA dictionary and its application in our analysis.

6. In the study design (Figure 1), it shows that there are 1535 ADEs reporting docetaxel as the primary suspected drug and 4347 events of docetaxel induced ADEs. Please clarify the differences between the two searches and why only the former was included in the final analysis.

Response:

Thank you for your thoughtful question regarding the differences between the two searches and the inclusion of 1,535 reports in the final analysis. We would like to clarify that the 4,347 represents the total number of adverse events (AEs) recorded within these 1,535 reports. Since a single patient (or report) may experience multiple adverse events (for example, both "fever" and "neutropenia"), the total number of adverse events is much higher than the number of reports.

Therefore, it is not the case that only the 1,535 reports were included in the final analysis. The entire analysis was based on the 4,347 adverse events documented in these 1,535 reports. Frequency analysis and signal detection for adverse event types (such as Preferred Term (PT) terms) were performed using the total number of adverse events.

We hope this clarification resolves the confusion and provides a clearer understanding of our analysis approach.

7. It seems the figure legends for Figures 1 to 3 are missing from the main manuscript. Also, Figure 3 is not cited in the text. Please add the missing information.

Response:

Thank you for your valuable feedback regarding the missing figure legends for Figures 1 to 3 and the issue with Figure 3 not being cited in the text. We have reviewed the manuscript and made the necessary revisions:

Figure Legends: We have now added the legends for Figures 1 to 4 to ensure that all figures are clearly described.

Figure 3 Citation: We have ensured that Figure 3 is properly cited in the manuscript, particularly in the "3.6 Adverse Event Occurrence Time and Weibull Distribution Analysis" section.

Figure Legends:

Fig 1. Flowchart demonstrating the adverse event analysis process for Docetaxel using the FDA Adverse Event Reporting System (FAERS) database.

Fig 2. Proportion of adverse events by system organ class associated with Docetaxel. Explanation: The 'Investigations' category refers to adverse events related to abnormal laboratory test results. 'Social circumstances' refers to social factors that influence the patient's health status or reporting behavior.

Fig 3. Time to onset of Docetaxel-induced adverse events.

Fig 4. Cumulative incidence of Docetaxel-related adverse events over time.

We hope these revisions address your comment and improve the clarity of the manuscript.

8. Please enlarge the font size and graph for Figure 2. It is difficult to view the variable labels as presented.

Response:

Thank you for your valuable feedback regarding the font size and clarity of the graph in Figure 2. In response to your comment, we have made the following revisions to improve the presentation of the figure:

Font Size and Boldness: We have increased the font size and bolded the variable labels in Figure 2 to ensure they are clearer and more legible.

Image Resolution: We have also enhanced the resolution of Figure 2 to improve the overall clarity of the graph.

We believe these changes will improve the readability and overall quality of the figure.

9. For Table 1, change the label for missing data from “Miss” to “Unknown.”

Response:

Thank you for your helpful suggestion regarding the label for missing data in Table 1. We have made the requested change and updated the label from "Miss" to "Unknown." The revised table has been re-uploaded accordingly.

We appreciate your attention to detail and believe this modification improves the clarity of the manuscript.

10. Please reformat Supplementary Tables 3 to 9 in the Supplementary Material file to ensure the table columns are wide enough to accommodate the numbers to prevent wrapping and allow clarity in viewing.

Response:

Thank you for your valuable suggestion regarding the formatting of Supplementary Tables 3 to 9. We have adjusted the table columns to ensure they are wide enough to accommodate the numbers without wrapping, improving clarity and readability. The revised tables have been re-uploaded accordingly.

We appreciate your attention to detail and believe these changes enhance the presentation of the supplementary material.

Reviewer 2

1. Figure 1 could be refined, as it currently includes some dashes and symbols that seem out of place or may confuse the reader.

Response:

Thank you for your constructive feedback regarding Figure 1. We have reviewed the figure and deleted the dashes and symbols that seemed out of place and could potentially confuse the reader. The revised figure has been re-uploaded accordingly.

We appreciate your suggestion, and we believe these changes enhance the clarity and presentation of the figure.

2. In the adverse effects graph, the categories "Investigations" and "Social circumstances" are unclear. It would be helpful to provide a brief explanation of what these terms refer to in this context.

Response:

Thank you for your valuable feedback regarding the clarity of the categories "Investigations" and "Social circumstances" in the adverse effects graph. In response to your suggestion, we have added explanations for these terms in the figure legend. Specifically:

The 'Investigations' category refers to adverse events related to abnormal laboratory test results.

The 'Social circumstances' category refers to social factors that influence the patient's health status or reporting behavior.

We believe this addition will help clarify these categories for the reader.

---

## [Decision Letter · Decision Letter 1]

24 Aug 2025

Assessing the real-world safety of Docetaxel for non-small cell lung cancer: insights from a comprehensive analysis of FAERS data

PONE-D-25-01841R1

Dear Dr. Wang,

We’re pleased to inform you that your manuscript has been judged scientifically suitable for publication and will be formally accepted for publication once it meets all outstanding technical requirements.

Kind regards,

Guocan Yu

Academic Editor

PLOS ONE

Additional Editor Comments (optional):

Reviewers' comments:

Reviewer's Responses to Questions

**Comments to the Author**

1. If the authors have adequately addressed your comments raised in a previous round of review and you feel that this manuscript is now acceptable for publication, you may indicate that here to bypass the “Comments to the Author” section, enter your conflict of interest statement in the “Confidential to Editor” section, and submit your "Accept" recommendation.

Reviewer #1: All comments have been addressed

Reviewer #2: All comments have been addressed

2. Is the manuscript technically sound, and do the data support the conclusions?

Reviewer #1: Yes

Reviewer #2: Yes

3. Has the statistical analysis been performed appropriately and rigorously? 

Reviewer #1: Yes

Reviewer #2: Yes

4. Have the authors made all data underlying the findings in their manuscript fully available?

Reviewer #1: Yes

Reviewer #2: Yes

5. Is the manuscript presented in an intelligible fashion and written in standard English?

Reviewer #1: Yes

Reviewer #2: Yes

6. Review Comments to the Author

Reviewer #1: Authors have satisfactorily addressed all of my comments and concerns. However, one minor detail concerning the alleged correction of missing information in Table 1 from "Miss" to "Unknown" or "NA" (not available) is not seen in the revised manuscript.

Reviewer #2: Congratulations. I believe the authors have successfully addressed the reviewers' comments and incorporated the suggested improvements. This revised version is well-executed, and I am confident that readers will find it both informative and engaging

7. PLOS authors have the option to publish the peer review history of their article (what does this mean? ). If published, this will include your full peer review and any attached files.

**Do you want your identity to be public for this peer review?** For information about this choice, including consent withdrawal, please see our Privacy Policy .

Reviewer #1: **Yes: ** David W Chang

Reviewer #2: **Yes: ** Borja Aguinagalde

---

## [Editor Report · Acceptance letter]

PONE-D-25-01841R1

PLOS ONE

Dear Dr. Wang,

I'm pleased to inform you that your manuscript has been deemed suitable for publication in PLOS ONE. Congratulations! Your manuscript is now being handed over to our production team.

Kind regards,

on behalf of

Dr. Guocan Yu

Academic Editor

PLOS ONE